# Protein Concentration Affects the Food Allergen γ-Conglutin Uptake and Bacteria-Induced Cytokine Production in Dendritic Cells

**DOI:** 10.3390/biom13101531

**Published:** 2023-10-16

**Authors:** Giuditta C. Heinzl, Danny Blichfeldt Eriksen, Peter Riber Johnsen, Alessio Scarafoni, Hanne Frøkiær

**Affiliations:** 1Department of Food, Environmental and Nutritional Sciences (DeFENS), University of Milan, Via Celoria 2, 20133 Milan, Italy; giuditta.heinzl@unimi.it (G.C.H.); alessio.scarafoni@unimi.it (A.S.); 2Department of Veterinary and Animal Sciences, University of Copenhagen, Ridebanevej 9, 1871 Frederiksberg, Denmark; qlm234@alumni.ku.dk (D.B.E.); peterriber@sund.ku.dk (P.R.J.)

**Keywords:** food allergen, γ-conglutin, dendritic cells, uptake, aggregation, microbial stimuli

## Abstract

γ-Conglutin (γ-C) from lupin seeds has been identified as a potent allergen with cross reactivity to peanuts. Here, we investigated how γ-C affected the response in bone marrow-derived dendritic cells (DCs) to bacterial stimuli. γ-C enhanced *L. acidophilus* NCFM (LaNCFM)-induced IL-12, IL-10, and IL-23 dose-dependently. In contrast, together with *E. coli* Nissle or LPS, γ-C reduced the production of IL-12 but not of IL-23 and IL-10. Enzyme-hydrolyzed γ-C also enhanced LaNCFM-induced IL-12 and IL-23 production. All preparations induced ROS production in the DCs. The mannose receptor ligands mannan and dextran and the clathrin inhibitor monodansylcadaverine partly inhibited the endocytosis of γ-C. Kunitz trypsin inhibitor and the scavenger receptor ligand polyG also enhanced LaNCFM-induced IL-12, indicating the involvement of receptors other than C-type lectin receptors. The endocytosis of labeled γ-C increased dose-dependently by addition of unlabeled γ-C, which coincided with γ-C’s tendency to aggregate. Taken together, γ-C aggregation affects endocytosis and affects the cytokine production induced by gram-positive and gram-negative bacteria differently. We suggest that γ-C is taken up by the same mechanism as other food proteins but due to aggregation is present in higher concentration in the DCs. This could influence the resulting T-cell response in a microbial stimuli-dependent way.

## 1. Introduction

Lupine belongs to the legume family together with soy, peanut, and other types of legumes and can be grown with high yield in Europe [1]. Like soy and peanuts, lupine elicits allergic reactions in some individuals and has shown cross-reactivity with peanut proteins [2,3,4]. In particular, lupine β- and δ-conglutins cross-react with the peanut proteins Ara h 1 and Ara h 2, respectively, while α- and γ-conglutins cross-react with Ara h 2 and Ara h 3, respectively [5]. In 1994, the first case of an allergic reaction caused by the ingestion of spaghetti enriched with lupine flour was reported [6]. The first recorded case of anaphylactic response dates back to 1999 [7]. With the increase in the use of lupine flour as a food ingredient, the number of adverse reactions has increased over the years [8,9,10]. Numerous clinical studies have been carried out [2,11,12,13] together with the implementation of methodologies to detect the presence of lupine in food and understand the structure of allergenic proteins [14,15,16,17]. For this reason, the knowledge of the mechanisms of action causing the onset of the allergic reaction is of particular interest and may lead to the design of more effective food technologies and food compositions capable of mitigating allergic reactions. The ability to induce an allergy has also been shown in the choleratoxin (CT)-induced food allergy mouse model [18], in which a response highly similar to that found in humans was induced [19]. Both in humans and in the CT mouse model, one protein, γ-conglutin (γ-C), stands out by eliciting a strong IgE response upon ingestion. Γ-C is a glycoprotein of 46 kDa consisting of two subunits, 29 and 17 kDa, respectively, linked together with a single disulfide bond and accounting for only 5% of the total protein in *Lupinus albus* [20]. The mature native protein undergoes pH-dependent quaternary organization in solution, forming an oligomeric assembly as a hexamer at pH 7.5 and a monomer when the pH shifts to acidic values [21].

In order to induce an antibody response towards a protein, the protein must be endocytosed by dendritic cells, and the cell should concurrently receive a microbial signal in order to upregulate MHC-II and costimulatory receptors and stimulate naïve T-cells [22]. The microbial signal is usually provided by intact bacteria or bacterial compounds. The type of microbial signal is expected to play a key role in the type of response that is elicited [23,24]. Human as well as murine dendritic cells were demonstrated to endocytose glycosylated allergens [25,26]. Salazar et al. [27] demonstrated that the mannose receptor (MR) facilitated uptake of mannosylated glycoproteins and inhibited LPS-induced IL-12, as well as the Th-1 cell-produced IFN-γ. We have previously shown that mannan-induced MR endocytosis enhanced the IL-12 production induced by gram-positive bacteria while reducing the IL-12 production induced by gram-negative bacteria [28]. Hence, endocytosis through such C-type lectin receptors may modulate the cytokine response, depending on the microbial signal. To improve our understanding of which factors may be involved in the antigenic and allergenic properties of γ-C, we investigated the effect of endocytosis of intact and enzyme-digested γ-C and how the protein modulated bacterially induced cytokine response in bone marrow-derived dendritic cells (DCs).

## 2. Materials and Methods

### 2.1. Preparation of γ-Conglutin Hydrolysate

To evaluate the effect of not fully digested proteins on DCs, purified γ-C [21] was hydrolyzed with bovine trypsin or with pepsin and pancreatin (Sigma-Aldrich, St. Louis, MO, USA) in an enzyme/protein ratio of 1:30, as previously described [29]. In brief, the γ-C protein was hydrolyzed with trypsin in ammonium bicarbonate (pH 7.5) for 45 min or with pepsin in 1 mM HCl (pH 3.0) for 45 min followed by hydrolysis with pancreatin in ammonium bicarbonate (pH 7.5) for 45 min. The hydrolysis was stopped by heating for 30 min at 65 °C, and samples were freeze-dried for subsequent use.

The purity of the γ-C protein was assessed through SDS-PAGE analysis (Appendix A). γ-C digested samples were analyzed by RP-HPLC using a SIMMETRY300 C18 (5 µm) (4.6 mm × 250 mm) column (Waters, Sesto San Giovanni, Italy) fitted on a chromatographic apparatus (Waters) composed of two 510 HPLC Pumps, a 717plus Autosampler, and a 996 Photodiode Array Detector. The mobile phase flux was 0.8 mL/min, mixing solutions A (TFA 0.1% in water) and B (TFA 0.1% in ACN) as follows: 2 min isocratic 100% solution A, 50 min linear gradient to 25% solution A, and 75% solution B. Peaks were detected at 220 and 280 nm.

### 2.2. Generation of Murine Dendritic Cells

Bone marrow-derived dendritic cells (DCs) were prepared from C57BL/6 mice (C57BL/6NTac, Taconic, Lille Skensved, Denmark) as previously described [27]. Cells were cultivated in RPMI 1640 medium (Sigma-Aldrich, St. Louis, MO, USA) containing 10% (vol/vol) heat-inactivated fetal calf serum (FCS) and supplemented with penicillin, streptomycin, glutamine, 2-mercaptoethanol, and murine granulocyte-macrophage colony-stimulating factor (GM-CSF). The cells were incubated for 8 days at 37 °C in a humidified 5% CO_2_ atmosphere. On day three and on day six, a new medium with GM-CSF was added, while on day eight, non-adherent cells were harvested.

### 2.3. Stimulation of DCs with Proteins, Bacteria, LPS, and Cytokine Production

Immature DCs were harvested and seeded in a 48-well plate at a concentration of 2 × 10^6^ cells/mL. Cells were supplemented with different concentrations of intact protein or peptides or polyG (Sigma-Aldrich) and incubated for 30 min at 37 °C in 5% CO_2_. Then, lipopolysaccharide (LPS) (100 ng/mL), *Escherichia coli* Nissle 1917 (MOI 10), or *Lactobacillus acidophilus* NCFM (LaNCFM) (MOI 10) was added. Cytokine secretion from DCs was measured in cell culture supernatants after 20 h of incubation, using duoset ELISA (R&D systems, Minneapolis, MN, USA) for murine IL-10 (DY417), IL-12p70 (DY419), IL-23 (DY1887), and IFN-β (DY8234) according to the manufacturer’s instructions.

### 2.4. Preparation of Labeled Proteins and Bacteria

For endocytosis experiments, intact γ-C, Kunitz trypsin inhibitor (KTI, Sigma-Aldrich T-9003), and LaNCFM were labeled with Alexa Fluor 647-conjugated succinimidyl ester (SE-AF647) (Molecular Probes, Eugene, OR, USA). The protein and bacteria were solubilized in carbonate buffer (pH 8.5) at a concentration of 10 mg/mL. To 100 µL of the protein or bacteria, 5 µL of SE-AF647 was added, and the mixture was incubated at 4 °C for 1 h in the dark with agitation. In order to remove unbound compounds from the proteins, a PD-10 Desalting Column (GE Healthcare) was used. The sample was eluted with sterile PBS (Gibco™, Thermo Fisher, Waltham, MA, USA), and the presence of the protein was assessed with a BioPhotometer. In the bacteria preparation, unbound compounds were removed by centrifugation. The contents of endotoxin in all protein and nucleotide preparations at a concentration of 100 µg/mL were determined by a Pierce™ Chromogenic Endotoxin Quant Kit (ThermoFisher Scientific, Roskilde, Denmark).

### 2.5. Protein and Bacteria Uptake by DCs

To assess uptake of protein by flow cytometry, Alexa Flour-647 conjugated proteins and bacteria were used. BMDCs were harvested and seeded at a concentration of 2 × 10^6^ cells/mL in a 96-well round bottom plate. Pre-treatment of cells with mannan (30, 100, 300 µg/mL), FITC-dextran (500 µg/mL) or unlabeled protein (10, 50, 100 µg/mL), monodansyl cadaverine (MDC) (100 µg/mL), or cytochalasin D (CytD) (0.5 µg/mL) (all from Sigma-Aldrich) or unlabeled protein was carried out for 30 min at 37 °C in a humidified 5% CO_2_ atmosphere. After the incubation, labeled protein was added, and cells were further incubated for 30 min under the same conditions. The cells were washed twice with sterile PBS and fixed with 1% formaldehyde in PBS. Uptake of protein and bacteria was analyzed by flow cytometry on a FacsCantoII system (BD Biosciences, San Jose, CA, USA), based on counting 20,000 cells, and mean fluorescence intensity (MFI) was used to determine the effect of different stimuli. All data are from conditions tested in triplicate. Data analysis was performed using FlowJo 10.6.2 software (Ashland, OR, USA).

### 2.6. Assessment of Protein Aggregation by Light Scattering

Analysis of the aggregation status of labeled and not labeled γ-C and KTI (2 mg/mL) was performed by using the serial system composed by a HPLC pump (515 HPLC pump, Waters), a SEC column (Superose 12, 10/300, GE Healthcare, Milan, Italy), a UV spectrophotometer (Dual λ absorbance detector 2487, Waters), a multi-angle light scattering device (DAWN HELEOS, Wyatt, Santa Barbara, CA, USA), provided with a fast photon counter (QELS), and a differential refractometer (Waters), as previously described in Caparo et al. [21]. Two hundred µL of samples, previously centrifuged to remove the insoluble fraction, were injected with PBS (pH 7.5) as mobile phase, at a flow rate of 0.5 mL/min. Molar masses were calculated by means of the Astra V software vs. 5.3.4.20 (Wyatt), using a dn/dc value of 0.185 for γ-C samples.

### 2.7. Measurement of ROS Formation

DCs (2 × 10^6^ cells/mL) were seeded in 96-well plates, incubated with 5 μM carboxy-H_2_DCFDA (Thermo Fisher Scientific, Waltham, MA, USA), and treated with 100 μg/mL of intact protein for 4 h at 37 °C, 5% CO_2_. The oxidation was quantified using flow cytometry (BD FacsCanto II system, BD Biosciences, San Jose, CA, USA) by measuring the FITC intensity. PBS and LPS-stimulated cells were used as negative and positive controls, respectively. All data are from conditions tested in quadruple.

### 2.8. Statistical Analysis

Data were analyzed using GraphPad Prism 9 and are presented as means ± standard errors. Results were tested with one-way ANOVA, as the data were normally distributed. Differences between control and treatment groups were considered significant when *p* < 0.05.

## 3. Results

### 3.1. DCs Stimulated with E. coli Nissle 1917 or L. acidophilus NCFM Are Modulated Differently by γ-C

To investigate the influence of purified γ-C protein (Appendix A) on the cytokine production of diversely microbially stimulated immature dendritic cells, DCs were treated with increasing concentrations of γ-C 30 min prior to stimulation with the gram-positive bacterium LaNCFM, the gram-negative *E. coli* Nissle 1917, or LPS from *E. coli*. In immature DCs, γ-C alone induced a minor production of the cytokines studied, albeit to a lesser extent than the microbial-induced production (Figure 1).

All three microbial stimuli-induced IL-12, IL-10, and IL-23 concentrations were around 300 pg/mL. Treatment with γ-C (10–100 µg/mL) prior to stimulation with *L. acidophilus* NCFM led to a three-fold increase of the IL-12, Figure 1A. The concentration of IL-23 showed a similar increase upon treatment with γ-C, while the increase in IL-10 was two-fold (Figure 1A). In contrast, γ-C added prior to stimulation with LPS or *E. coli* Nissle 1917 showed a decreasing effect on the IL-12 production at the highest concentration of γ-C (100 µg/mL), did not affect the IL-23 production, and showed a moderate dose-dependent increase in IL-10 production (Figure 1B,C).

### 3.2. Both Intact and Degraded Protein Holds IL-12 Enhancing Capacity in L. acidophilus NCFM Stimulated DC

To further study the effects of γ-C on the gram-positive LaNCFM-induced cytokine response, we investigated if both the native structure of γ-C and enzymatically digested γ-C were capable of enhancing the cytokine response. We preincubated the DCs with γ-C equivalents of the hydrolyzed protein (same concentration as of the intact protein) before adding LaNCFM (Figure 2).

The elution profiles obtained by RP-HPLC analysis of γ-C samples showed the intact protein as a peak at about 35 min. When the native γ-C was incubated with trypsin, only a limited and incomplete cleavage of the protein took place. In contrast, the pepsin-and pancreatin-treated sample showed a higher extent of degradation with a strong reduction in intact γ-C (Figure 2A). When treating DCs with increasing concentrations of intact or digested γ-C protein for 30 min prior to stimulation with LaNCFM, the enzyme-treated γ-C samples gave rise to an enhanced production of IL-12 and IL-23 in levels comparable to that of intact γ-C (Figure 2B), indicating that the enhancement is not dependent on the intact native protein. Γ-C is a glycoprotein containing different sugar moieties, including N-acetyl-glucosamine and mannose residues [30]. We have previously demonstrated that carbohydrates such as mannan and β-glucans can enhance the IL-12 and IFN-β production induced by the gram-positive bacteria in DCs [28,31]. Accordingly, we compared the effect of adding mannose and γ-C prior to stimulation with LaNCFM over a period of post-stimulation (Figure 3A). Compared to LaNCFM-stimulated DCs, pre-treatment with mannan led to increased and prompter IL-12 production, while the IL-10 remained largely the same. Pre-treatment with γ-C showed a more potent increase in IL-12 production with no major change in IL-10, and albeit differences in the degree of enhancement, the kinetics in the production were highly similar, indicating that mannan and γ-C affect the cells by the same mechanism. We have previously shown that LaNCFM induces a potent IFN-β response that in turn leads to the majority of the produced IL-12 [32,33]. The IFN-β production followed the course of IL-12, indicating that the enhancement in IL-12 production is due to enhanced IFN-β production (Figure 3A).

To study the putative role of the mannose receptor in the endocytosis of γ-C, DCs were added, either mannan or dextran, both shown to bind to MR [34,35,36]. This was followed by the addition of AF657-labeled γ-C (Figure 3B). Compared to cells only, added AF657-labeled γ-C both mannan (i) and dextran (ii) reduced the intensity of AF-657 in DCs as measured by flow cytometry. Together, these results suggest that the mannose receptor may play a role in the enhanced IL-12 response by γ-C. We cannot, however, exclude that other endocytosing receptors on DCs, e.g., scavenger receptors (ScRs), may contribute to the endocytosis of γ-C.

### 3.3. A Non-Glycosylated Protein Is as Potent as γ-C and Mannan in Enhancing L. acidophilus NCFM-Induced IL-12 and IFN-β

To investigate if a non-glycosylated protein would also have the capacity to enhance the *L. acidophilus*-induced IL-12 and IFN-β, DCs were preincubated with γ-C or Kunitz trypsin inhibitor (KTI) from soybeans (Mw 20.1 kDa) before stimulation with LaNCFM (Figure 4A). Compared to γ-C, at lower concentrations, KTI gave rise to a higher enhancement of IL-12 and IFN-β, but at the highest concentration (100 µg/mL), the IL-12 and IFN-β concentrations dropped, indicating that an upper level can be reached. The purity of the KTI protein was assessed in SDS-PAGE along with γ-C in Appendix A. As for γ-C, KTI did not significantly affect the *E. coli* Nissle 1917-induced cytokine response. Some ScRs have been shown to be capable of binding the nucleoside polyguanidine (polyG) [37,38,39]. To assess the putative role of ScRs in the enhanced IL-12 response by γ-C, we tested if pre-treatment with polyG also led to enhanced IL-12 and IFN-β production in LaNCFM-stimulated DCs. Like the two proteins, polyG induced a dose-dependent enhancement of IL-12 and IFN-β (Figure 4a). Several ScRs depend on clathrin-mediated endocytosis (e.g., MR, SCRs group A) [28,40]. To assess if the receptors involved in the IL-12 enhancement were dependent of clathrin-coated pits pinocytosis, we used MDC to inhibit the pinocytosis (Figure 4B). The enhancement in IL-12 and IFN-β production by both proteins was reduced, but not to the level of stimulation with only LaNCFM (Figure 4B). Adding the MDC inhibitor to cells prior to the addition of AF657-labeled γ-C likewise gave rise to a reduction in uptake of protein (Figure 4C). By using AF657-labeled LaNCFM, we could rule out that addition of the protein affected the endocytosis of bacteria. Together, these results indicate that multiple receptors may bind and endocytose γ-C and other proteins, leading to enhanced IL-12 response.

### 3.4. Addition of γ-C or KTI Induces ROS Production

We have previously found that mannan’s interaction with MR induces enhanced Il-12 induction from gram-positive bacteria through increased formation of ROS [28]. To investigate if γ-C and KTI enhance the IL-12 production by inducing endosomal ROS production, we assessed the capacity of the two proteins and polyG to induce ROS formation (Figure 5A,B). Both proteins and polyG induced increased oxidation of 6-carboxy-2′,7′-dichlorodihydrofluorescein diacetate (carboxy-H_2_DCFDA) in a dose-dependent manner, indicating increased ROS production, and furthermore enhanced the LPS-induced ROS production (Figure 5B). Inhibition of NADPH oxidase assembly and activation by addition of apocynin strongly reduced the IL-12 production; the IFN-β production was almost completely abrogated, while the IL-10 production remained the same (Figure 5C). As many protein preparations are contaminated with LPS [41,42], and LPS induces ROS and cytokine production in DCs [43], we determined the endotoxin content in the protein preparations, in Table 1. The endotoxin level ranged from 0 (under detection limit) in the polyG to 20 EU/mg. Thus, although the protein preparation contained some endotoxin, the levels in all samples were well below the LPS concentration used for cell stimulation. Taken together, these data indicate that the proteins hold ROS-inducing capacity beyond the effect caused by the LPS contamination.

### 3.5. In High Concentration, γ-C Is Endocytosed in Higher Concentrations Due to Aggregation

When increasing concentrations of unlabeled γ-C were added to compete with AF657-labeled γ-C, an unexpected dose-dependent increase in cells with a higher fluorescence intensity appeared (Figure 6). Addition of increasing concentrations of unlabeled KTI did marginally affect the uptake of labeled γ-C in the cells. In contrast, when increasing concentrations of unlabeled KTI were added together with labeled KTI, a reduction in AF657 intensity was observed. Increasing γ-C concentrations together with labeled KTI did not alter the intensity or number of labeled cells significantly (Figure 6). The cells with the highest fluorescence intensity also exhibited a more granular phenotype, indicating the presence of large amounts of protein in the cells. As we saw a higher uptake of labeled γ-C when adding increasing concentrations of γ-C but not of KTI, we speculated if γ-C tended to form aggregates in a concentration-dependent way. To distinguish between receptor-mediated pinocytosis and macrocytosis, we treated the cells with Cytochalasin D (CytD) alone or together with non-labeled protein prior to adding labeled protein (Figure 7A,B). CytD had some reducing effect on the uptake of labeled γ-C. Incubating with unlabeled γ-C resulted in a doubling of the labeled γ-C in the cells, which was almost fully abrogated by CytD. Likewise, if unlabeled KTI was added, uptake of labeled γ-C increased; however, together with CytD, it decreased to below half of the uptake, and after treatment with CytD alone, a slight reduction in the γ-C was seen, indicating a higher γ-C uptake by micropinocytosis but also some competition in receptor-mediated uptake between γ-C and KTI. The uptake of labeled KTI was reduced to the same extent by CytD and CytD + KTI (Figure 7A,B). The different patterns in uptake of γ-C and KTI indicate that γ-C forms larger aggregates in higher concentration. To investigate if labeling could influence the propensity of γ-C to form soluble aggregates (supramolecular interactions) other than the quaternary structure, a light scattering analysis was performed (Figure 7C). Chromatograms showed that the labeled γ-C had a slightly lower level of propensity (about 30%) to form aggregates compared to unlabeled γ-C (the peak at 15 min corresponding to Mw > 2000 kDa). On the other hand, peaks at 20 and 23 min, corresponding to the hexameric and dimer forms, respectively, were not affected. In contrast, KTI and labelled KTI showed no differences, and KTI did not aggregate. As shown in Figure 7D, at low concentration (0.2 mg/mL), γ-C does not form aggregates, while at 1 mg/mL and even more at 2 mg/mL (cfr. Figure 7C,D), the aggregates emerge in the chromatograms. Since the cell experiments were performed at up to 0.3 mg/mL protein (or up to 0.6 mg/mL in competitive labeled/unlabeled protein experiments), the observed increased protein uptake (Figure 6) is likely due to the presence of supramolecular aggregates. It is therefore evident from the experiments conducted that the ability of γ-conglutin to form aggregates leads to a higher uptake by DCs.

Hence, regarding the uptake of γ-C and KTI by DCs, the two proteins show discrete uptake patterns depending on the concentration, and abortion of the macropinocytosed γ-C demonstrated some competition between the two proteins.

## 4. Discussion

Despite much concern, it is still not fully clear why some plant proteins are allergenic, and others are not. Endocytosis of antigens by dendritic cells is essential for the induction of T-cell activation and polarization playing a key role in the immune responses to food proteins. This is of major importance for our understanding of why some proteins in some individuals give rise to an allergy. Here, we investigated the interaction of a dietary allergenic protein, γ-C, with DCs. We found that, depending on the microbial stimuli, γ-C enhances or reduces the induced Th polarizing cytokine response. In particular, in conjunction with γ-C, the gram-positive probiotic bacteria *L. acidophilus* NCFM, known to stimulate the immune system towards a Th1 (non-allergenic) inducing IL-12 response [33,44], γ-C more than doubled the IL-12, IL-23, and IL-10 production in dendritic cells. In contrast, together with the gram-negative bacteria *E. coli* Nissle 1917, γ-C led to decreased IL-12 and IL-23 production. This may add to our understanding of how the microbiota influences the propensity to develop an allergy. Such an effect was previously demonstrated to be mediated by mannan and β-glucan [31] and involves the C-type lectin receptors MR and dectin-1. In particular, the mannose receptor is a distinctive protein that interacts with glycans. Its structure comprises three distinct types of domains: firstly, a single cysteine-rich glycan-binding domain located at the N-terminus; secondly, a fibronectin type II domain; and lastly, a sequence of eight consecutive CTLD domains. The cysteine-rich domain specifically recognizes sulfated glycans, while the CTLD domains collectively exhibit the capability to bind to various monosaccharides, including mannose, fucose, N-acetylglucosamine, and glucose, and larger glycans, especially the mannose-rich ones [45]. Salazar et al. [27] demonstrated that certain glycoproteins reduced the LPS-induced IL-12 response, but to our knowledge, the IL-12 enhancing effect of a protein on the IL-12 inducing effect of a gram-positive bacteria has not been demonstrated before. γ-C is a glycoprotein, and both mannose and dextrose reduced the uptake of γ-C. The glycosylation of γ-C has been extensively studied by Schiarea et al. [30] and Czubinski et al. [46], and up to 20 different glycosidic structures have been identified. The most abundant glycans observed contain mannose, fucose, xylose, and N-acetylglucosamine residues with *GlcNAc1-2Man3XylGlcNAc2Fuc* as composition. However, peaks in smaller quantities containing glucose or galactose residues have also been identified [46]. It is worth noting that its allergenic properties could potentially be influenced by the type of bound carbohydrate. The ability of MR to internalize proteins is supported by Autenrieth et al. [47], who showed that DCs were able to take up a lower amount of ovalbumin mediated by MR. This indicates a role of MR in the endocytosis of γ-C and a putative role in the enhanced IL-12 production; however, it does not exclude that other non-C-type lectin receptors are involved in the endocytosis of γ-C. On the contrary, the fact that the non-glycosylated KTI as well as polyG are likewise endocytosed by an MDC-dependent mechanism, leading to enhanced IL-12 upon stimulation with the gram-positive bacterium LaNCFM, establishes that other non-C-type lectin receptors are involved. PolyG is a known ligand of scavenger receptor group A [48,49], and ligation of the scavenger receptors was demonstrated to induce ROS production [37,38,39], thus supporting that various scavenger receptors may be involved in the endocytosis and IL-12 enhancing effect of proteins. The C-type lectin receptors have recently been included in the group of scavenger receptors, and upon ligation, internalization of the receptors leads to ROS production [31]. Here, we also found an increased ROS production in the DCs by the addition of γ-C and KTI as well as PolyG, indicating a role of one or more non-C-type lectin scavenger receptors in the uptake of the proteins and concomitant enhancement of IL-12.

Even though all protein preparations induced ROS and enhanced the production of IL-12 in the DCs per se, caution should be taken in concluding that the proteins alone are responsible for this, as we found that both the γ-C and KTI protein preparation were significantly contaminated with endotoxin (LPS), which holds ROS-inducing activity [41,42] and the ability to induce IL-12 in DCs [50]. Addition of the proteins alone to the DCs gave rise to induction of some IL-12 production. Hence, we cannot exclude that contaminating LPS contributes to the effect seen by adding the protein. Some facts, however, point towards the idea that most of the observed effect is caused by the protein per se. Firstly, we have previously studied the effect of preincubating DCs with LPS prior to stimulation with LaNCFM and found that LPS led to a reduction of induced IL-12 [50]. With addition of the proteins, we observed a highly significant increase in the IL-12 production that did not correspond to the level induced by the protein alone; hence, the endotoxin levels in the added protein seem too low to circumvent the activity of the proteins. Secondly, we could not measure any endotoxins in the PolyG preparation. As PolyG induced the same response in the DCs as the proteins (ROS production, enhanced IL-12 and IFN-β production in LaNCFM-stimulated cells), it seems likely that γ-C and KTI induce similar effects through the interaction with one or more scavenger receptors. Lastly, we have previously found that LPS increased the uptake of labeled LaNCFM by DCs but reduced the IL-12 production by LaNCFM [50], while in the present study, we did not find an increased LaNCFM uptake by addition of the protein. Together, this supports that the primary cause of the protein-induced enhancement of IL-12 and IFN-β and the induction of ROS production are caused by the ligation and endocytosis of the proteins by scavenger receptors or other receptors that employ clathrin-mediated uptake.

After all, although of great interest as regards how protein antigens and microorganisms mutually affect the induced immune response towards proteins, we did not reveal any difference between the food-allergen γ-C from lupin seeds and KTI from soybeans with low allergenicity. To search for other putative differences between the two proteins, using AF657-labeled proteins, we studied the uptake of γ-C and KTI in DCs and revealed that the two proteins showed distinct concentration-dependent endocytosis patterns. While the uptake of labeled γ-C increased with higher concentrations of unlabeled γ-C, the uptake of labeled KTI decreased with increasing concentration of unlabeled KTI. This discrepancy may be explained in at least two ways; either the two proteins employ different receptors with different uptake kinetics, or, more likely, γ-C but not KTI forms aggregates in a concentration-dependent way. To pursue the hypothesis that the discrepancy between the uptake patterns of the two proteins is caused by the aggregation of γ-C, we analyzed the aggregation of γ-C and KTI, both labeled and unlabeled, and found that KTI retained its monomeric form, while most of the γ-C attained a hexameric structure, as well as larger aggregates above 2000 kDa. Molecules and particles up to around 100 nm can be internalized by receptor-mediated pinocytosis, but in dendritic cells, also by macropinocytosis, where the dendritic cells take up big gulps of the surrounding liquid, including the molecules and particles that are present in the liquid [51]. Particles much larger than 200 nm can be endocytosed by the latter mechanism. When we inhibited macropinocytosis in the DCs, we could see a minor reduction in the uptake of labeled γ-C and a more pronounced reduction in the uptake of labeled KTI. This supports that a proportion of the protein in solution is taken up by macropinocytosis but also that receptor-mediated pinocytosis plays a key role in the uptake. Most importantly, when unlabeled KTI was added together with CytD and either labeled γ-C or KTI, the uptake of labeled protein was significantly reduced, indicating that the two proteins at least partly compete for the same receptors. This strongly supports that the increase in endocytosis of labeled γ-C is caused by aggregates formed of labeled and unlabeled γ-C, allowing a higher proportion of the labeled γ-C simultaneously by the receptor.

The observation that a higher concentration of γ-C (and to some extent, other proteins) leads to the formation of aggregates of the γ-C and accordingly results in endocytosis of a higher number of the protein molecules has to our knowledge not been shown before. It was previously suggested that the propensity to form aggregates may constitute an intrinsic property of allergens [52]. The presented results suggest that aggregated proteins are endocytosed in higher amounts, which may be degraded by the antigen-presenting cell and presented by a higher number of MHC molecules. This is, however, purely speculative, as we did not investigate the presentation of γ-C peptides on MHC; neither did we show if the putative increased uptake affects the number or efficiency of Th1 cell activation. Anyway, uptake of higher numbers of antigen molecules in antigen-presenting cells per se may not lead to induction of an allergy. According to Costa et al. [52], protein aggregation has been demonstrated to contribute to increased allergenicity of 2S albumins but to reduce allergenicity for legumins and cereal prolamins. We previously demonstrated that mice ingesting soy protein generate an antibody response with reactivity towards the quaternary structure of glycinin, possibly reflecting that antibodies primarily are generated towards multimeric forms of proteins [53]. This measured antibody response was not an IgE response, underscoring that it is not the concentration of endocytosed antigen per se that determines the immune polarization. However, together with our finding that the type of microbial stimuli (as shown here by a gram-positive and a gram-negative bacteria) influences how the antigen type and concentration and the microbial stimuli mutually determine the cytokine production in dendritic cells, our demonstration of how protein concentration can affect aggregation and endocytosis may add to our understanding of intrinsic and environmental factors involved in the onset of allergy.

## 5. Conclusions

The understanding of why some food proteins are allergenic and others are not is still lacking. Here, we studied the plant allergen γ-C from lupin compared to the non-allergenic Kunitz trypsin inhibitor in how it interacts with dendritic cells. Both proteins were endocytosed by clathrin-dependent receptor mediated uptake and modulated the cytokine production induced by gram-positive and gram-negative bacteria differently. γ-C degraded by digestive enzymes exhibited the same effect as the intact protein. Due to aggregation of γ-C, increasing protein concentration led to more endocytosed γ-C, as determined by fluorescence. A similar increase was not observed for KTI. This is the first study to demonstrate the distinct immunomodulatory effect of γ-C that depends on whether it is combined with stimulating gram-positive or gram-negative bacteria. We suggest that the tendency to aggregation increases the likelihood of antigen presentation, and in combination with the specific microbial stimuli, this may play a role in the resulting response to a food protein.

## Figures and Tables

**Figure 1 biomolecules-13-01531-f001:**
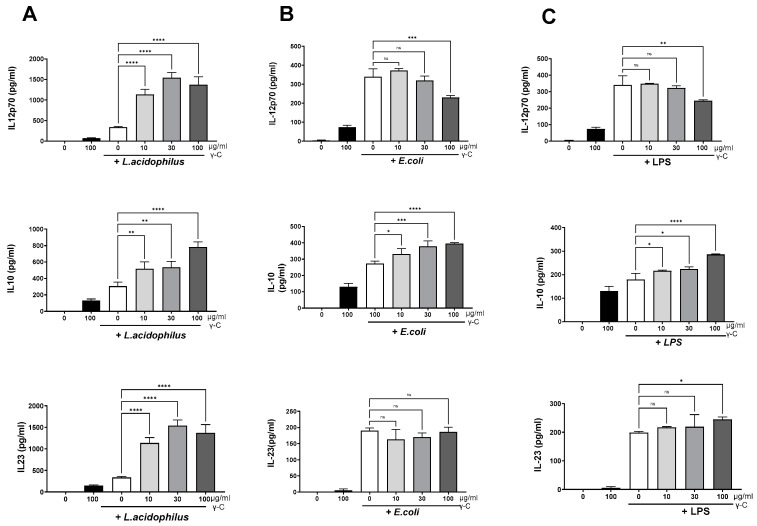
Effects of preincubation with intact γ-conglutin (γ-C) on the *L. acidophilus* NCFM (**A**), *E. coli* Nissle 1917 (**B**), and LPS (**C**) induced production of IL-12, IL-10, and IL-23 by DCs. DCs were pretreated with the protein at 10, 30, and 100 µg/mL for 30 min then treated with the indicated microbial stimuli for 20 h. The culture supernatants were collected, and the concentrations of cytokines were quantitated by ELISA. * *p* < 0.05, ** *p* < 0.01, *** *p* < 0.001, **** *p* < 0.0001, ns: not significant.

**Figure 2 biomolecules-13-01531-f002:**
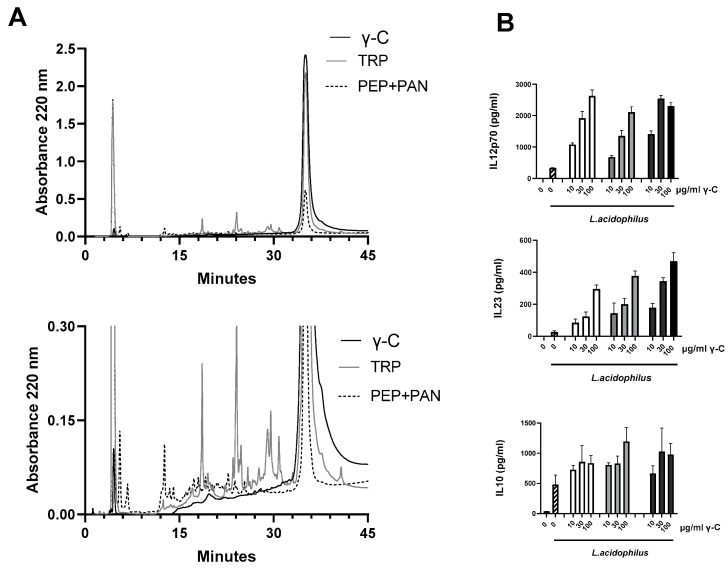
The effect of trypsin-treated γ-C and pepsin-pancreatin-treated γ-C on LaNCFM-induced cytokine response. (**A**): Upper part: RP-HPLC chromatogram showing the profile of undigested γ-C (black line), trypsin (TRP, grey line) and pepsin and pancreatin (PEP + PAN, dotted line) digested γ-C, respectively. The peak at 36 min corresponds to intact protein. Lower part: enlarged part of the chromatogram showing minor peaks. (**B**): DCs were added, increasing concentrations of native γ-C (white bars), γ-C treated with trypsin (grey bars), or γ-C treated with pepsin and pancreatin (black bars) and incubated for 20 h before harvest of supernatant. The concentrations of IL-12, IL-23, and IL-10 were determined by ELISA.

**Figure 3 biomolecules-13-01531-f003:**
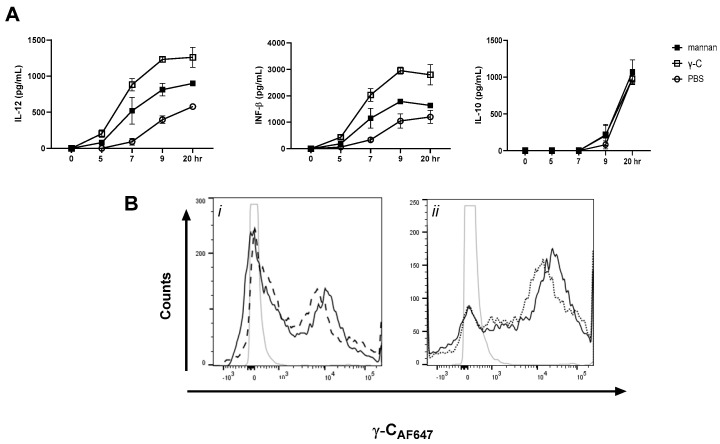
Mannose receptor contributes to the uptake and the γ-C enhanced cytokine response towards LaNCFM. (**A**): The concentrations of IL-12, IL-10, and IFN-β in the supernatants of DCs pre-incubated with mannan (100 µg/mL), γ-C (100 µg/mL), or PBS for 30 min prior to stimulation with *L. acidophilus* NCFM for 0, 5, 7, 9, and 20 h. (**B**): AF657 positive cells after pretreatment of DCs with AF657-labeled γ-C (solid line) in competition with (i): dextran (stippled line) or (ii): mannan (dotted line), measured by flow cytometry. Grey lines are unstained cells as control.

**Figure 4 biomolecules-13-01531-f004:**
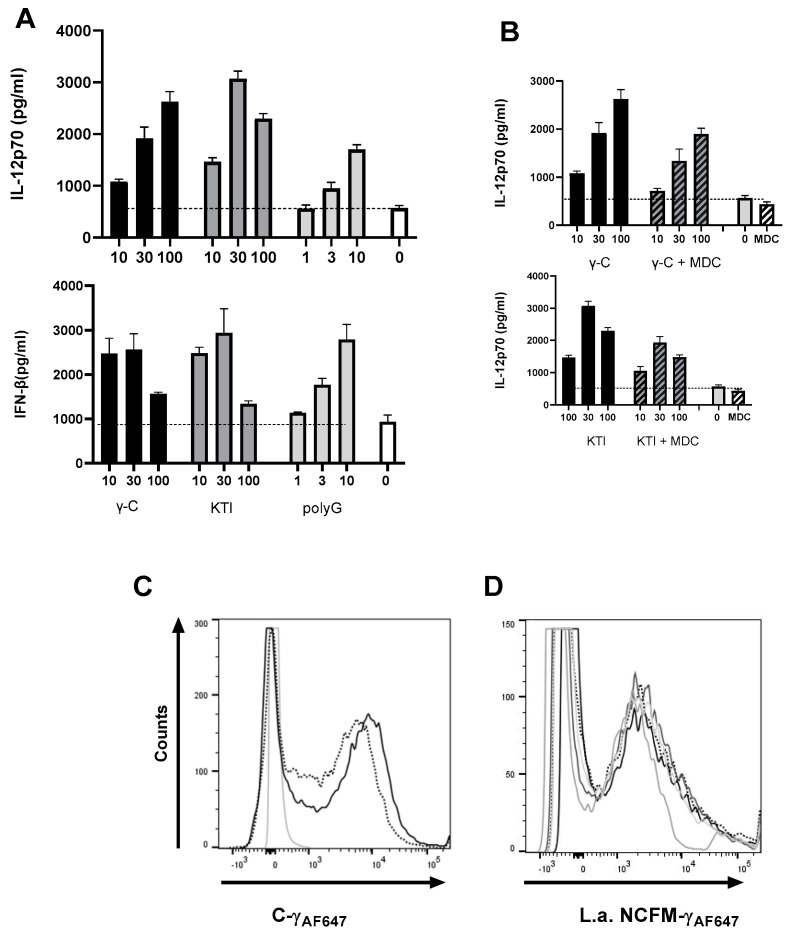
KTI and polyG as good as γ-C in enhancing LaNCFM-induced IL-12. (**A**): Effect of increasing concentrations of γ-C and KTI (10, 30, and 100 µg/mL) and of PolyG (1, 3, and 10 µg/mL) on *L. acidophilus* NCFM-induced IL-12 and IFN-β production. The stippled line indicates the cytokine level after stimulation with LaNCFM with no protein added. (**B**): Effect of monodansyl cadaverine (MDC) on the C-γ- or KTI-enhanced IL-12 response in bmDCs stimulated with *L. acidophilus* NCFM. MDC was added to the cells 60 min and C-γ or KTI (10, 30 and 100 µg/mL) 30 min prior to stimulation with *L. acidophilus* NCFM. The stippled line indicates the cytokine level after stimulation with LaNCFM with no protein or MDC added. (**C**): Effect of MDC on the uptake of fluorescently labeled γ-C (γ-C_AF657_) in bmDCs. MDC (dotted) or media (solid) was added to bmDCs 30 min prior to γ-C_AF657._ After an additional 30 min of incubation, the intensity of fluorescence in each cell was measured by flow cytometry. (**D**): γ-C and KTI do not affect uptake of *L. acidophilus* NCFM. The proteins (100 µg/mL) ((γ-C (black), KTI (dark grey), CytD (grey), Nystatin (light grey), media (dotted)) were added 30 min prior to addition of AF567-labeled *L. acidophilus* NCFM (L.a.NCFM_AF657_). Cells were analyzed by flow cytometry after 1 h of incubation.

**Figure 5 biomolecules-13-01531-f005:**
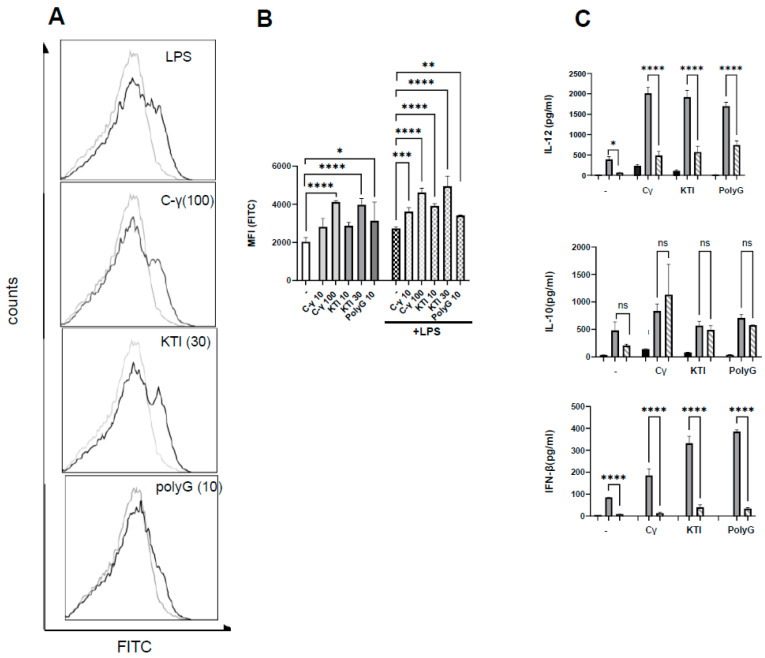
ROS production is induced by γ-C, KTI, and PolyG and may be involved in the enhanced IL-12 and IFN-β. (**A**): Histograms of cells incubated with carboxy-H_2_DCFDA prior to treatment with LPS (100 ng/mL), γ-C (100 µg/mL), KTI (30 µg/mL), or PolyG (10 µg/mL) for 30 min or left untreated. (**B**): Bar graph showing the mean fluorescence I = intensity (MFI) of cells incubated with carboxy-H_2_DCFDA and pre-treated with γ-C (10 or 100 µg/mL, light grey), KTI (10 or 30 µg/mL, gray), or PolyG (10 µg/mL, dark gray) with (hatched bars) or without addition of LPS (1 µg/mL). (**C**): Cytokine concentration of supernatant from DCs treated with (gray hatched bars) or without (gray bars) apocynin, then pre-incubated with γ-C, KTI, or polyG and stimulated *L. acidophilus* NCFM. Cells with added proteins without stimulation with *L. acidophilus* NCFM are shown as black bars. * *p* < 0.05, ** *p* < 0.01, *** *p* < 0.001, **** *p* < 0.0001, ns: not significant.

**Figure 6 biomolecules-13-01531-f006:**
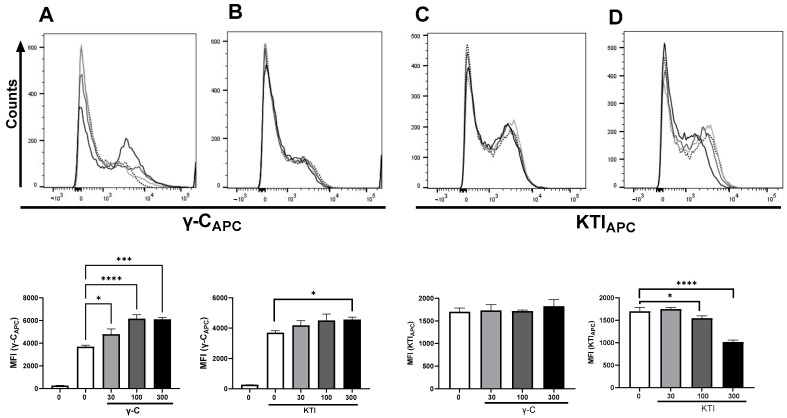
At high γ-C concentrations, more labeled γ-C is endocytosed. Labeled γ-C (**A**,**B**) or labeled KTI (**C**,**D**) was added alone or together with increasing concentrations of unlabeled γ-C (**A**,**D**) or unlabeled KTI (**B**,**C**). In each panel, the figures in (**A**) depict the histogram of each combination at the indicated concentration of unlabeled protein; 0 µg/mL (dotted), 30 µg/mL (light gray), 100 (dark gray), and 300 µg/mL (black), and in (**B**): the average MFI of each concentration (0–300 µg/mL) of added unlabeled protein to the indicated labeled protein. * *p* < 0.05, *** *p* < 0.001, **** *p* < 0.0001.

**Figure 7 biomolecules-13-01531-f007:**
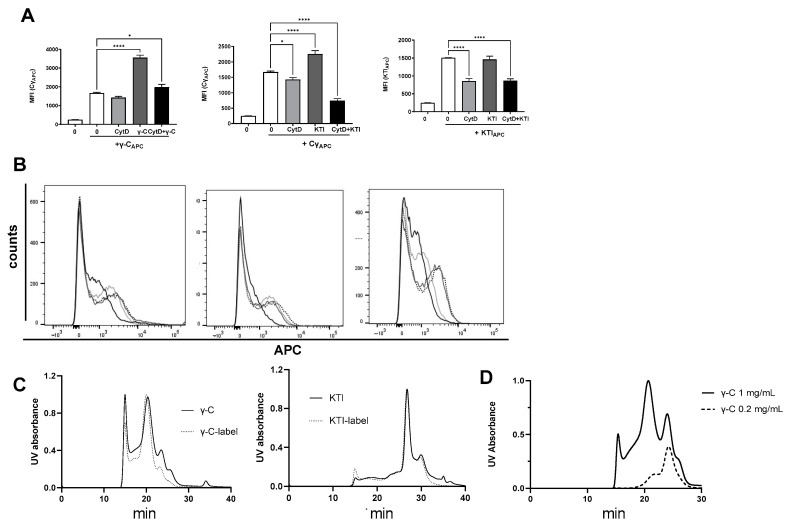
At high protein concentration, γ-C forms aggregates. AF657-labeled γ-C was added to DCs treated for 30 min with media (white or dotted line), cytochalasin D (CytD) (gray), unlabeled γ-C or KTI (light gray), or both (dark gray), and the uptake of AF657 was evaluated by flow cytometry depicted as mean fluorescence intensity (MFI), *n* = 4 (**A**) and histograms (**B**). (**C**): chromatograms of solutions of γ-C and labeled γ-C (2 mg/mL, left) and of KTI and AF657-labeled KTI (2 mg/mL, right). (**D**): Chromatography of γ-C in solution at 0.2 and 1 mg/mL. The signal at 280 nm is shown. * *p* < 0.05, **** *p* < 0.0001.

**Table 1 biomolecules-13-01531-t001:** Contamination of protein preparations by endotoxins.

Preparation	Endotoxin (EU/mg)
Conglutin-γ	20
Kunitz trypsin inhibitor	0.5
PolyG	nd
Pep-pan conglutin-γ	30
Trp Conglutin-γ	30

nd: Not detectable; Pep-pan: pepsin and pancreatin treated; Trp: trypsin treated. 1 EU/mL~0.2 ng/mL.

## Data Availability

The data presented in this study are available on request from the corresponding author. The data are not publicly available due to the nature of the data which cannot be uploaded in a single dataset.

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
