# Peer review of "Protein Concentration Affects the Food Allergen γ-Conglutin Uptake and Bacteria-Induced Cytokine Production in Dendritic Cells"

_biomolecules, 2023, doi:10.3390/biom13101531_

Round 1

Reviewer 1 Report

Manuscript by Heinzl et al. describes studies of lupin γ-conglutin influence on cytokine production in dendritic cells. The manuscript focuses on a topic that is interesting not only from the perspective of allergy, but also from the perspective of interaction of human immune system with different external factors. The manuscript is generally well written; however, there are two points that should be addressed by the authors. First, it seems that there is a fragment of the text that is missing (see page 4, line 166), and there is no continuation of the text from page 4 to page 5. Please clarify this issue. The second problem is related to the experiment with hydrolyzed γ-conglutin. The authors use bovine trypsin or pepsin and pancreatin in an enzyme/protein 1:30 ratio. However, there is no information whether the enzymes alone were used in controlled experiments. This issue should be addressed and clarified. For example, one may expect that such controls will be shown in Figure 2a.

In addition, there are multiple minor issues that should be corrected/addressed.

1. Page 1, lines 11-12

Please clarify what you meant by γ-conglutin being cross-reactive to peanut. As I understand, there is a particular protein from peanuts that is similar to the one from lupin.

2. The authors are inconsistent with notation of Latin organisms’ names. In several places the names are in italics, or there is no space between genus and species. Please correct these issues. Similarly, when a particular organism’s name appears for the first time in the text, the whole name should be provided.

3. Page 3, line 141

Please clarify what “was washed twice at 400xG” means.

4. For example, page 4, line 148

The authors are not consistent with unit notation and sometimes there is “ml” and sometimes “mL” used in the text. Please unify the notation through the whole text and figures.

5. Figures’ notation is inconsistent, as sometimes capital letters or small cases are used to mark different parts of the figures. For example, see Figure 1 vs. Figure 2.

6. Figure 2a & Figures C & D

There is no unit provided for time.

7. In some sections of the manuscript, a different font size is used. It is not clear whether this was done on purpose. For example, please see page 8, line 247; page 11, lines starting from 315, etc.

8. Quality of figures showing flow cytometry results should be improved. For example, there is a manually added Y-axis description (Counts) that does not cover the original label.

The manuscript is generally well written.

Author Response

We thank the reviewer for the interest expressed on out research topic.

First, it seems that there is a fragment of the text that is missing (see page 4, line 166), and there is no continuation of the text from page 4 to page 5. Please clarify this issue. 

Answer: The reviewer is right; the layout was wrong. The issue has been corrected.

The second problem is related to the experiment with hydrolyzed γ-conglutin. The authors use bovine trypsin or pepsin and pancreatin in an enzyme/protein 1:30 ratio. However, there is no information whether the enzymes alone were used in controlled experiments. This issue should be addressed and clarified. For example, one may expect that such controls will be shown in Figure 2a.

Answer: We decided to not include this control in figure 2A as the signal at 220 of the peptides released by enzyme autolysis was barely detectable at the shown chromatograms scaling. This because the enzymes are only one thirtieth of the total proteins and the estimated percentage of autolysis of the enzymes is about 27% as also reported by Menard et al. 2023 https://doi.org/10.1016/j.foodres.2023.113242.

  1. Page 1, lines 11-12

Please clarify what you meant by γ-conglutin being cross-reactive to peanut. As I understand, there is a particular protein from peanuts that is similar to the one from lupin.

Answer: A citation has been added in order to clarify which lupin proteins cross-react with peanut proteins (lines 32-33).

  1. The authors are inconsistent with notation of Latin organisms’ names. In several places the names are in italics, or there is no space between genus and species. Please correct these issues. Similarly, when a particular organism’s name appears for the first time in the text, the whole name should be provided.

Answer: The Latin organisms’ names have been carefully revised according to reviewer suggestion. All the text has been checked.

  1. Page 3, line 141

Please clarify what “was washed twice at 400xG” means.

Answer: This should be changed to: The pellet was washed in sterile PBS followed by centrifugation at 400G two times. However, as this experiment has been omitted, l.  140-149 has accordingly been deleted.

  1. For example, page 4, line 148

The authors are not consistent with unit notation and sometimes there is “ml” and sometimes “mL” used in the text. Please unify the notation through the whole text and figures.

Answer: The paper has been revised for unit notation inconsistency.

  1. Figures’ notation is inconsistent, as sometimes capital letters or small cases are used to mark different parts of the figures. For example, see Figure 1 vs. Figure 2.

Answer: Figures’ notation has been revised and corrected.

  1. Figure 2a & Figures C & D

There is no unit provided for time.

Answer: Unit (minutes) has been added for time.

  1. In some sections of the manuscript, a different font size is used. It is not clear whether this was done on purpose. For example, please see page 8, line 247; page 11, lines starting from 315, etc.

Answer: The reviewer is right; the size and character of the manuscript has been corrected.

  1. Quality of figures showing flow cytometry results should be improved. For example, there is a manually added Y-axis description (Counts) that does not cover the original label.

Answer: Figures have been corrected and improved.

Reviewer 2 Report

This study addressed immunological property of an allergen, γ-conglutin from lupin seeds. Such information promotes to understand whether stimulatory effect of allergens on the innate immunity is a factor to explain its potential allergenicity. Therefore, the rational is understandable. However, this reviewer concerns following points.

(1) Please show the purity of γ-conglutin, at least by SDS-PAGE.

(2) The figure presentation is very unfriendly to the reviewer. The caption is too small. In several figures, there are no indication of concentration units. Fig. 3: which figures are A, B, and C? Please improve the quality or figure presentation and legends.

(3) LPS concentration in the Conglutin samples is high, around 2000 to 4000 pg/mg of protein (20 EU/mg). The author should remove the LPS using Triton114 and then assess the immuno-stimulatory effect of the conglutin sample.

https://journals.plos.org/plosone/article?id=10.1371/journal.pone.0173778

(4) Fig.3: The expression levels of the mannose receptors look comparable among the group. It does not support the involvement of the receptor in the binding to γ-conglutin. If the authors would like to show the involvement of the receptor, gene know down, or knock out of the target receptor should be considered.

(5) Fig.7: It is still not clear whether aggregation of conglutin contributes to its immunological property.

(6) Does the allergen enhance expression of costimulatory molecules and T cell stimulatory effect of DC? Induction of IL-12 would lead to Th1 differentiation, which rather inhibits allergic response.

(7) The authors described that γ-conglutin decreased IL-12 in co-stimulation with gram-negative bacteria E.coli. However, the reduction is observed only at a high concentration of γ-conglutin, and appears to be simply due to over stimulation. In a lower concentration, there is a trend that co-stimulation with γ-conglutin and E.coli promotes IL-12 production.

(8) Overall, it is not clear what is a novel finding in this study, and how the data can explain for the potential allergenicity of γ-conglutin.

(9) It is also important to include glucan information of γ-conglutin if the stimulatory effect of this allergen is due to the presence of glucan.

The authors should ask professional English editing.

Author Response

Reviewer 2

(1) Please show the purity of γ-conglutin, at least by SDS-PAGE.

Answer: SDS-PAGE of purified γ-conglutin protein has been added in supplementary material as Supplementary Figure S1.

(2) The figure presentation is very unfriendly to the reviewer. The caption is too small. In several figures, there are no indication of concentration units. Fig. 3: which figures are A, B, and C? Please improve the quality or figure presentation and legends.

Answer: Figures have been improved and corrected.

(3) LPS concentration in the Conglutin samples is high, around 2000 to 4000 pg/mg of protein (20 EU/mg). The author should remove the LPS using Triton114 and then assess the immuno-stimulatory effect of the conglutin sample.

Answer: We are aware of the high LPS contamination in the conglutin samples, but we were concerned that the treatment of the conglutin with TritonX114 would cause deaggregation of the protein. AS we added 10-100 ug protein/sample, this corresponded a lps concentration of 40-400pg/ml, which may result in a minor stimulation of the cells. In e.g. figures 1,2 and 6 we show the effect of adding the protein alone without any microbial stimuli and from these it is evident that although the protein per se induces a low level of IL-12, the enhancement in IL-12 seen when the protein is added together with L.acidophilus NCFM is far higher .

As we discuss in line 400-422., our data shows that LPS as such does not cause the enhancement of the IL-12 and IFNb production (Boye et al., Immunology 2016) and moreover that even though KTI  and PolyG contained markedly lower or no  LPS, these molecule exhibit similar IL-12 and IFNb enhancing capacity.   Thus, the LPS contamination of conglutin leads to little IL-12 production per se but this LPS cannot be the cause of the enhancement of the L.acidophilus induced IL-12 response.

(4) Fig.3: The expression levels of the mannose receptors look comparable among the group. It does not support the involvement of the receptor in the binding to γ-conglutin. If the authors would like to show the involvement of the receptor, gene know down, or knock out of the target receptor should be considered.

Answer: We agree that this experiment does not in a convincing way support the involvement of the MR. Accordingly, this information has been removed from the manuscript so that the analysis suggested by the reviewer could be carried out and presented in future manuscripts.

(5) Fig.7: It is still not clear whether aggregation of conglutin contributes to its immunological property.

Answer: A sentence has been added to the text to explain the role of aggregation (lines 344-345). Please also see the answer to point 6.

(6) Does the allergen enhance expression of costimulatory molecules and T cell stimulatory effect of DC? Induction of IL-12 would lead to Th1 differentiation, which rather inhibits allergic response.

Answer:  we have data that indicate an upregulation of CD83 and CD86 by C-g and KTI. However, due to the LPS contamination, we cannot be sure wether the upregulation is due to the presence of LPS or protein, or a combination, so we chose not to include the data in the manuscript.

 Our conclusion of the data in the manuscript is that :

1) the type of bacteria present together with an antigen is an important co-determinant of the polarization of the cytokine response (and thus, the immune response) induced. While it today is generally accepted that gram-positive and -negative bacteria influence the immune response differently, the novelty presented here is that we show  that the protein antigen may affect the microbially induced cytokine response; and

2)  compared to a non-allergenic protein, the C-g stands out by forming aggregates that leads to a greater uptake of C-g in dendritic. The amount of antigen is important for the type  and strenght of  immune response. In particular, the initial T-cell response magnitude and subsequent development and retention of memory T cells are directly related to the antigen dose. 

This has to our knowledge not been reported before and may – although we do not here demonstrate the direct causality - help explaining how an intrinsic  property of an allergen  (tendency to form aggregates leading to a higher uptake of the protein in the dendritic cell) and the specific microbial signal may determine the outcome: Th2 vs Th1 (allergy inducing  vs a non-allergenic response).

We have tried to make this clearer in the manuscript.

(7) The authors described that γ-conglutin decreased IL-12 in co-stimulation with gram-negative bacteria E.coli. However, the reduction is observed only at a high concentration of γ-conglutin, and appears to be simply due to over stimulation. In a lower concentration, there is a trend that co-stimulation with γ-conglutin and E.coli promotes IL-12 production.

Answer: In Figure 1 B the γ-conglutin (10 µg/mL) and E. coli co-stimulation causes an increase of IL-12 production that is not significant compared to the control. In the various repetitions of this experiment, we see minor variations in the responses but only addition of the high concentration of protein consistently gave rise to a reduction of IL-12, which very well may be due to over-stimulation. Compared to the enhancement of IL-12 seen when the proteins are added together with the gram-positive L.acidophilus NCFM  any up- or downregulated of the E.coli stimulated IL-12 seem of minor significance.

(8) Overall, it is not clear what is a novel finding in this study, and how the data can explain for the potential allergenicity of γ-conglutin.

Answer: The manuscript investigates for the first time the effect of the allergen γ-conglutin protein and its derived peptides on DCs. Moreover, the mechanisms of uptake and in particular the effect of protein uptake on the microbially induced cytokine production have never been investigated in this context. Together with the finding of the tendency of C-g to aggregate leading to uptake of a greater amount of antigen, which is known to influence the immune response, our data may help explaining why a protein like C-g but not KTI gives rise to allergy.

(9) It is also important to include glucan information of γ-conglutin if the stimulatory effect of this allergen is due to the presence of glucan.

Answer: References about γ-conglutin glycosylation have been added along with glucan potential role in allergy response (lines 379-382).

However, we want to stress that although our results support a role of the glucan part of C-g for uptake and influence on the cytokine production, we cannot conclude that this is only due to the glycosylation of C-g. Rather, the fact that a non-glycosylated  protein affet the cytokine pattern in a similar way indicates that the glycosylation  has only mine impact.

Round 2

Reviewer 1 Report

In the current version of the manuscript the authors addressed my major concerns. In the current version Figure S1 should be corrected. Namely the molecular weight standards should have information on a unit used, which most likely is kDa. Please also note that in such a case the numbers should have periods, and not commas. In addition, the authors should comment on a band that is observed at molecular weight of ~90 kDa. Is it γ-Conglutin dimer? Is this somewhat related to the aggregation process reported in the manuscript?

The manuscript is generally well written.

Author Response

The figure has been corrected according to reviewer’s suggestions and a comment has been added to explain the bands. The γ-Conglutin bands corresponds to the monomer (≈45 kDa) and dimer (≈90 kDa) and more associated forms, respectively, as the native protein has been shown to undergo pH-dependent association–dissociation equilibrium between monomer and polymer(up to 6 polypeptides associate). In contrast, the aggregation of the protein (that is, multiple associated polypeptides aggregate to form large aggregates) depends on the concentration and may too large to be assessed by SDS-PAGE.

Reviewer 2 Report

This reviewer still concerns following points.

(1)The figure presentation is still very unfriendly. Please see figure 1. The numbers are too small. There is no caption in the middle and right figures. Fig.5C: Please explain about black, gray, and bar graphs. Fig.7A: Caption of X axis is not readable. Fig. 7B: media (white) is invisible. There is no explanation about dot line. 

(2) Line 369: Please check if the word ovolactalbumin is correct.

(3)  If the LPS is not involved in the cytokine production by tested proteins, how KTI can induce the production and ROS in dendritic cells? KTI is non-glycosylated protein. In general, non-glycosylated proteins do not stimulate DCs. Can the author exclude contamination of other proteins, e.g. lectins, which exert DC activation. Please add SDS-PAGE data of not only Conglutin but also KTI. It is also important to include catalog number of product in Sigla Aldrich, as not all products are well purified in the vender.

(4) Line 346-347: The authors should describe glucan information of Conglutin more, and discuss which glucan could bind to mannose receptor or other C type lectin receptors.

Author Response

(1) The figure presentation is still very unfriendly. Please see figure 1. The numbers are too small. There is no caption in the middle and right figures. Fig.5C: Please explain about black, gray, and bar graphs. Fig.7A: Caption of X axis is not readable. Fig. 7B: media (white) is invisible. There is no explanation about dot line.

Answer: The figures and figure text have been improved according to the reviewer points.

(2) Line 369: Please check if the word ovolactalbumin is correct.

Answer: The reviewer is right, the correct word is ovoalbumin.

(3) If the LPS is not involved in the cytokine production by tested proteins, how KTI can induce the production and ROS in dendritic cells? KTI is non-glycosylated protein. In general, non-glycosylated proteins do not stimulate DCs. Can the author exclude contamination of other proteins, e.g. lectins, which exert DC activation. Please add SDS-PAGE data of not only Conglutin but also KTI. It is also important to include catalog number of product in Sigma Aldrich, as not all products are well purified in the vender.

Answer: We acknowledge that adding KTI to unstimulated immature DCs gives rise to a weak cytokine response, which we attribute to the LPS contamination. We measured an endotoxin content of 10 pg/mg KTI, which means that adding 10-100 µg/ml KTI to the cells corresponds to 0.1-1 pg/ml of LPS. This is well beyond the LPS concentration usually used for stimulation (we used 100 ng/ml LPS in fig.1). However, when adding KTI together with a Gram-positive bacteria we see an increase in the bacteria-induced IL-12 response that far exceeds the sum of IL-12 induced by the bacteria and KTI, respectively. This indicates that KTI in a hitherto undescribed way enhances the G+ bacteria induced IL-12.

We see the KTI induce ROS production to an extent that cannot be attibuted to the low LPS content. This was also a surprise to us, as we only expected carbohydrates and glycoproteins to be able to induce ROS. We have actually now tested many proteins of various origins, and independently of LPS contamination, we find that most are potent ROS inducers. However, the proteins per se do not stimulate the DCs to produce a cytokine response.

The catalogue number of KTI , Sigma Aldrich T9003, has been added in M&M and to the SDS-PAGE as requested by the reviewer (figure S1). The protein is not completely pure as it is possible to see from the gel. We know from previous studies that the major contaminant of this preparation is the Bowman Birk inhibitor.

(4) Line 346-347: The authors should describe glucan information of Conglutin more, and discuss which glucan could bind to mannose receptor or other C type lectin receptors.

Answer: A more detailed description of γ-conglutin glycosylation has been provided (lines 372-376), along with a mannose receptor ligands explanation (lines 361-367).
